# Nonlinear Model Predictive Control of Single-Link Flexible-Joint Robot Using Recurrent Neural Network and Differential Evolution Optimization

**Anlong Zhang** [1], **Zhiyun Lin** [2,*] , **Bo Wang** [1] **and Zhimin Han** [1]

[1] Artificial Intelligence Institute, School of Automation, Hangzhou Dianzi University, Hangzhou 310018, China; anlongzhang@hdu.edu.cn (A.Z.); wangbo@hdu.edu.cn (B.W.); hanzm@hdu.edu.cn (Z.H.)

[2] Department of Electrical and Electronic Engineering, Southern University of Science and Technology, Shenzhen 518055, China

[*] Correspondence: linzy@sustech.edu.cn

**Abstract:** A recurrent neural network (RNN) and differential evolution optimization (DEO) based nonlinear model predictive control (NMPC) technique is proposed for position control of a single-link flexible-joint (FJ) robot. First, a simple three-layer recurrent neural network with rectified linear units as an activation function (ReLU-RNN) is employed for approximating the system dynamic model. Then, using the RNN predictive model and model predictive control (MPC) scheme, an RNN and DEO based NMPC controller is designed, and the DEO algorithm is used to solve the controller. Finally, comparing numerical simulation findings demonstrates the efficiency and performance of the proposed approach. The merit of this method is that not only is the control precision satisfied, but also the overshoots and the residual vibration are well suppressed.

**Keywords:** flexible-joint robot; nonlinear model predictive control; differential evolution; recurrent neural network

## 1. Introduction

The control of the flexible-joint (FJ) robot has been a major research topic in the field of control theory and engineering for several decades [1–7]. The FJ robot benefits from the characteristic of inbuilt compliance that provides low output impedance, shock tolerance, and accurate force control [8]. Due to its benefits, the FJ robot has been widely used in many applications where robot interacts with environments or with humans, such as monopod hopping robots and exoskeletons [9]. However, the FJ robot is an under-actuated strong coupling nonlinear system [10]. The control of such a complex nonlinear system is a difficult task. Therefore, the goal of this study is to design a suitable controller for a single-link FJ robot, which can also be utilized for complicated nonlinear systems.

Model-free methods have been generally employed in the field of FJ robot control. The earliest influential control approach is the traditional proportional-derivative (PD) method with gravity compensation [11,12]. Different types of PD controllers have been proposed because of their simplicity and practicability [13–15]. To deal with the overshoots and residual vibration, a fuzzy proportional-integral-derivative (PID) controller was proposed to suppress the elastic torsional vibration [16], and a nonlinear state feedback controller was employed to suppress the residual vibration of FJ robot [10]. Although these techniques in the aforementioned references have acquired relatively excellent performance in the FJ robot control, there are still certain issues requiring attention. For instance, the parameters of model-free controllers must be adjusted according to the requirements of system performance. The control performance is sensitive to the controller parameters, and these parameters are complex to adjust. Unlike the model-free method, model predictive control (MPC) as a primary model-dependent control method is an efficient controller to handle the

performance requirements. For example, in [17,18], the MPC controller has been utilized for robot manipulator trajectory tracking. For high precision position tracking of the robot arm, a data-driven MPC method has been proposed which has shown performance improvement compared to the PID method [19]. The MPC controller has been performed well in process control, automotive systems, and robotics due to its advantages of versatility, robustness, and safety guarantees [20–24]. However, the major challenge in the MPC method is obtaining an accurate system dynamic model. As we all know, the model of the FJ robot system is prone to uncertainty disturbances, inaccurate parameters, and unknown model functions (e.g., the friction model), making MPC implementation difficult. The neural network (NN) approach has been widely used as a strong tool in dealing with uncertainties and unknown model functions. For instance, the NN method has been employed to approximate the friction model for implementing friction compensation [25,26], and it was applied to estimate the unknown model parameters and uncertainties for achieving adaptive control [27–29]. Nevertheless, to the best of our knowledge, few works use the NN method to approximate the FJ robot system dynamics.

Recently, the study of merging MPC and NN techniques has increased [30], in which the NN methods are utilized to deal with the difficulty of modeling system dynamics. For example, in [31], a deep recurrent neural network (RNN) MPC architecture has been established to slice foods. In [32–34], deep NN was used to approximate soft robot dynamics for implementing MPC. Besides, NN has been utilized to approximate the MPC laws in [35–39]. In robot system, the optimization problem of MPC is still the challenges due to the nonlinear dynamic model and other non-convex constraints [40]. In addition, the robot system often suffers the deadlock problem, which has been well investigated in [41–43]. A suitable method for solving the nonlinear MPC (NMPC) is differential evolution optimization (DEO). DEO is a heuristic method proposed by [44], which is effective for solving numerical optimization issues. DEO has been designed as a stochastic parallel direct search method, and there are many studies on parallel DEO [45–48]. The DEO algorithm has the benefit of being a global optimization technique that is simple to understand and implement, and has strong robustness and fewer parameters to be adjusted. Due to its advantages, DEO has been extensively investigated [49] and successfully applied in diverse fields, including robot manipulator systems [50], mobile robots [51–53], autonomous cars [54], spectrum sensing systems [55], and permanent magnet synchronous motor systems [56]. Although the integrating MPC and NN methods produced good results in robot applications, few researches are focusing on the position control of the FJ robot. On one hand, the FJ robot system dynamics are hard to obtain. Contrarily, the optimization process of NMPC is a nonlinear programming problem, which is tough to solve.

In this study, we present an RNN and DEO based NMPC approach for position control of a single-link FJ robot. The RNN is employed to approximate the system dynamics, and the DEO algorithm is applied to solve the NMPC controller. The key contributions of this research are summarized as follows:

- First, an RNN and DEO based NMPC method is proposed for the position control of a single-link FJ robot. The merit of this process is that not only is the control precision satisfied, but also the overshoots and the residual vibration is well suppressed.
- To overcome the difficulty of modeling, a simple three-layer RNN with leaky rectified linear units as an activation function (ReLU-RNN) is established to approximate the FJ robot dynamic model with satisfactory precision. Then, according to the RNN predictive model and MPC approach, an RNN and DEO based NMPC controller is designed, in which the DEO algorithm is applied to solve the controller.
- Finally, to demonstrate the efficiency and performance of this technique, some numerical simulation comparisons between our method and the PD method and the differential dynamic programming (DDP) [57] MPC approach have been established. Numerical simulation findings illustrate that the performance of this technique is superior to that of the PD and DDP MPC methods.

The remainder of this paper is organized as follows. In Section 2, the dynamic model of the single-link FJ robot, including the direct-current (DC) motor dynamics is established. In Section 3, the controller design is indicated. The numerical simulations are displayed in Section 4. Finally, the conclusion is given in Section 5.

## 2. Single-Link FJ Robot System Model

In this section, we establish the single-link FJ robot dynamic model with the DC motor dynamics being considered. The single-link FJ robot system, which can rotate in vertical plane, is shown in Figure 1.

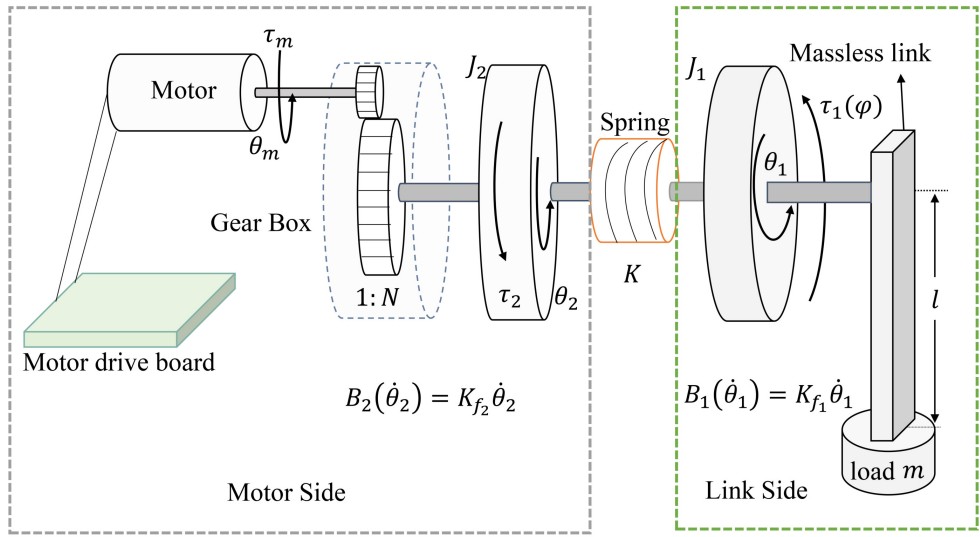

**Figure 1.** The architecture of single-link FJ robot system.

The system comprises two parts, as shown in Figure 1. The left part is the motor side, which includes a motor drive board, a DC motor, and a gear reduction box. The right part is the link side, which is composed of a massless link and a load. The two sections are linked by an elastic element, which is modeled as a linear spring. The FJ robot rotates in a vertical plane with the assumption that the elastic element can only deform in the direction of joint rotation [4]. The driving torque provided by the DC motor is $\tau_m$, and the gear reduction ratio is 1:N. The motor side torque is $\tau_2 = N\tau_m$. The stiffness of the linear spring is $K$. The angular position of the motor side is $\theta_2$, and the link side angular position is $\theta_1$. When the joint rotates, the joint can deform in the direction of rotation, and the torque can be represented by $\tau_1(\varphi) = K(\theta_2 - \theta_1)$, where $\varphi = \theta_2 - \theta_1$ denotes the deformation of a linear spring. $\dot{\theta}_1$ and $\dot{\theta}_2$ stand for the angular velocity of the link side and the motor side, respectively. Similarly, $\ddot{\theta}_1$ and $\ddot{\theta}_2$ symbolize the angular acceleration of the link side and the motor side, respectively. For the sake of simplicity, we presume the viscous damping on the motor side and the link side to be $B_1(\dot{\theta}_1) = K_{f_1}\dot{\theta}_1$ and $B_2(\dot{\theta}_2) = K_{f_2}\dot{\theta}_2$, where $K_{f_1}$ and $K_{f_2}$ denote the damping coefficient of the motor side and the link side, respectively. $G(\theta_1) = mgl\sin(\theta_1)$ represents gravity, where $m$ is the quality of the load, $g$ is the gravity acceleration, and $l$ is the length of the massless link. The rotary inertia of the link side and the motor side are $J_1$ and $J_2$, respectively. Then, based on the Euler–Lagrangian equations, the system dynamics is formulated as (1) [58]

$$\begin{cases} J_1\ddot{\theta}_1 + G(\theta_1) + B_1(\dot{\theta}_1) = \tau_1(\varphi), \\ J_2\ddot{\theta}_2 + \tau_1(\varphi) + B_2(\dot{\theta}_2) = \tau_2. \end{cases} \tag{1}$$

Since the motor is employed to actuate the system, the motor dynamics are also considered to institute the system dynamic model. The motor dynamics are depicted as (2)

$$\begin{cases} \tau_m = K_\tau i, \\ Ri + L\dot{i} + K_e \dot{\theta}_m = U_V, \end{cases} \tag{2}$$

where $K_\tau$ is motor torque coefficient, $i$ denotes motor armature current, $R$ represents armature circuit resistance, $L$ stands for armature circuit inductance, $K_e$ is back electromotive coefficient, $\dot{\theta}_m$ denotes the angular velocity of the motor rotor, and $U_V$ symbolizes motor armature voltage.

The torque produced by the motor is transmitted to the motor side using a gear reduction box as shown in Figure 1. We suppose that there is no transmission loss. Then, based on Equation (2), we attain

$$\begin{cases} \tau_2 = NK_\tau i, \\ \dot{\theta}_2 = \frac{1}{N} \dot{\theta}_m. \end{cases} \tag{3}$$

According to the above analysis, combining (1)–(3), the system dynamics including the motor dynamics can be described as (4)

$$\begin{cases} J_1 \ddot{\theta}_1 + mgl\sin(\theta_1) + K_{f_1}(\dot{\theta}_1) = K(\theta_2 - \theta_1), \\ J_2 \ddot{\theta}_2 + K(\theta_2 - \theta_1) + K_{f_2}(\dot{\theta}_2) = NK_\tau i, \\ Ri + L\dot{i} + NK_e \dot{\theta}_2 = U_V. \end{cases} \tag{4}$$

Let us define $x_1 = \theta_1$, $x_2 = \dot{\theta}_1$, $x_3 = \theta_2$, $x_4 = \dot{\theta}_2$, $x_5 = i$, $u = U_V$. Then, the system dynamics can be formulated by following state-space expression (5) and (6)

$$\dot{X}(t) = \begin{bmatrix} 0 & 1 & 0 & 0 & 0 \\ -\frac{K}{J_1} & -\frac{K_{f_1}}{J_1} & \frac{K}{J_1} & 0 & 0 \\ 0 & 0 & 0 & 1 & 0 \\ \frac{K}{J_2} & 0 & -\frac{K}{J_2} & -\frac{K_{f_2}}{J_2} & \frac{NK_\tau}{J_2} \\ 0 & 0 & 0 & -\frac{NK_e}{L} & -\frac{R}{L} \end{bmatrix} X(t) - \begin{bmatrix} 0 \\ \frac{mgl}{J_1}\sin x_1(t) \\ 0 \\ 0 \\ 0 \end{bmatrix} + \begin{bmatrix} 0 \\ 0 \\ 0 \\ 0 \\ \frac{1}{L} \end{bmatrix} u(t), \tag{5}$$

$$Y(t) = [1, 0, 0, 0, 0] X(t), \tag{6}$$

where $X(t) = [x_1(t), x_2(t), x_3(t), x_4(t), x_5(t)]^T$ denotes the system state vector, and $Y(t)$ symbolizes the system output. $u(t)$ stands for the control input of the system.

This model contains unmodeled parts, such as an accurate friction model, gear backlash, and mechanical transmission efficiency. Besides, precise model parameters are difficult to obtain. This type of nonlinear system is complex to control as the model is unknown. Thus, we present an RNN and DEO based NMPC method, which can be utilized for complicated nonlinear systems.

## 3. Controller Design

### 3.1. Nonlinear Model Predictive Control

Based on our system, the fundamental scheme of NMPC is detailed in this subsection. We utilize the discrete-time nonlinear autoregressive exogenous dynamic model to represent the system state-space Equation (5), which is capable of predicting future states for long-time series. At time step $k$, the state $X(k + 1)$ is predicted by (7)

$$X(k + 1) = f_p(X_k, U_k), \tag{7}$$

where $X_k = [X^T(k), X^T(k - 1), ..., X^T(k - d_x + 1)]$ represents system state time series from $k$th time step through $k - d_x + 1$th time step, correspondingly, $U_k = [u(k), u(k - 1), ..., u(k -$

$d_u + 1)]$ depicts the control input time series. $d_x$ and $d_u$ stand for the length of time series of system state and control input, respectively. $f_p(\cdot)$ signifies a nonlinear function.

For long-time series prediction, the predicted system state is transmitted into $X_k$ recurrently, for example (8)

$$X(k + 2) = f_p(X_{k+1}, U_{k+1}). \tag{8}$$

At $k + 1$th time step predicted system state $X(k + 1)$ is transmitted into $X_k$, and system state time series is updated as $X_{k+1} = [X^T(k + 1), X^T(k), ..., X^T(k - d_x + 2)]$. Similarly, exogenous control input is transmitted into $U_k$ and the control input time series is updated as $U_{k+1} = [u(k + 1), u(k), ..., u(k - d_u + 2)]$. This formula is not only useful for establishing NMPC controller but also convenient for approximating the model using NN.

Then, we consider the discrete-time nonlinear system (7) to express the MPC scheme. Equation (7) including constraints can be rewritten as (9)

$$\begin{aligned}\hat{X}(k + 1) &= f_n(X_k, U_k), \\ X_k &\in \mathcal{X}, \quad k = 0, 1, ..., N \\ U_k &\in \mathcal{U}, \quad k = 0, 1, ..., N\end{aligned} \tag{9}$$

where $\mathcal{X} \subset \Re^5$ symbolizes system state vector constraints, and $\mathcal{U} \subset \Re$ denotes control input constraints. $f_n(\cdot)$ stands for the nonlinear function, which is approximated by ReLU-RNN. $N$ is the prediction horizon.

A nonlinear MPC controller works by minimizing the performance criterion such as (10)

$$U^\star(k) = \underset{[u_0^\star, u_1^\star, ..., u_{N-1}^\star]}{\arg\min} J(\boldsymbol{X}(k), U(k)), \tag{10}$$

where $\boldsymbol{X}(k) = [X_k, X_{k+1}, ..., X_{k+N-1}]$ and $U(k) = [u_0, u_1, ..., u_{N-1}]$ symbolize the system state information and control input to be optimized, respectively. The cost function is denoted by $J(\boldsymbol{X}(k), U(k))$. $U^\star(k) = [u_0^\star, u_1^\star, ..., u_{N-1}^\star]$ signifies the optimized control input series. Each state-input pair satisfies Equation (9) with constraints. When the control input series are optimized, only the first term $u_0^\star$ is applied to the system until the next time step, and the system state measurements are updated at the next time step. Then, the optimization procedure is repeated at each time step, which runs as a closed-loop.

In the field of robot control, it is very important to realize accurate position control, speed control and torque control. In practical applications, the accuracy of position control will directly affect the performance of the robot. When performing position control, the residual vibration is easy to be inspired due to the existence of elastic elements [10]. Therefore, position control is the most important and more challenging in FJ robot control. In this study, in order to achieve accurate position control, the position variable is selected as the control objective of the NMPC controller, so that we design the quadratic cost function as (11)

$$J(\boldsymbol{X}(k), U(k)) = \alpha \sum_{j=0}^{N-1} \left[\hat{x}_1(k + j + 1) - x_1^{ref}(k + j + 1)\right]^2 + \beta \sum_{j=0}^{N-1} [u(k + j + 1) - u(k + j)]^2, \tag{11}$$

subject to the terminal constraint (12)

$$\hat{X}(k + N) = 0, \tag{12}$$

where $\hat{x}_1(k + j + 1)$ denotes predicted position state and $x_1^{ref}(k + j + 1)$ represents the reference trajectory. $u(k + j)$ stands for the system control input at time step $k + j$. Due to the constraints of the control input in the real system, this term is introduced into the objective function as adjustment. $\alpha$ and $\beta$ are the penalty coefficients of the performance criterion and control input, respectively.

We conclude from the above analysis that this technique is flexible as we have the option to design the cost functions for various objectives. For example, we can execute the velocity and torque control by simply varying the cost functions. However, implementing an NMPC controller is rather challenging, and there are two main problems in designing the controller. The first is how to create an accurate predictive model, and the second is how to solve the optimization problem successfully. We utilize a ReLU-RNN to approximate the system dynamic model in Section 3.2 to overcome the difficulty of nonlinear system modeling. The DEO algorithm, which will be described in Section 3.3 is used to optimize the control inputs.

### 3.2. Dynamics Model Approximation Using ReLU-RNN

We approximate the discrete-time dynamic model (7) by utilizing a simple three-layer ReLU-RNN, which is displayed in Figure 2.

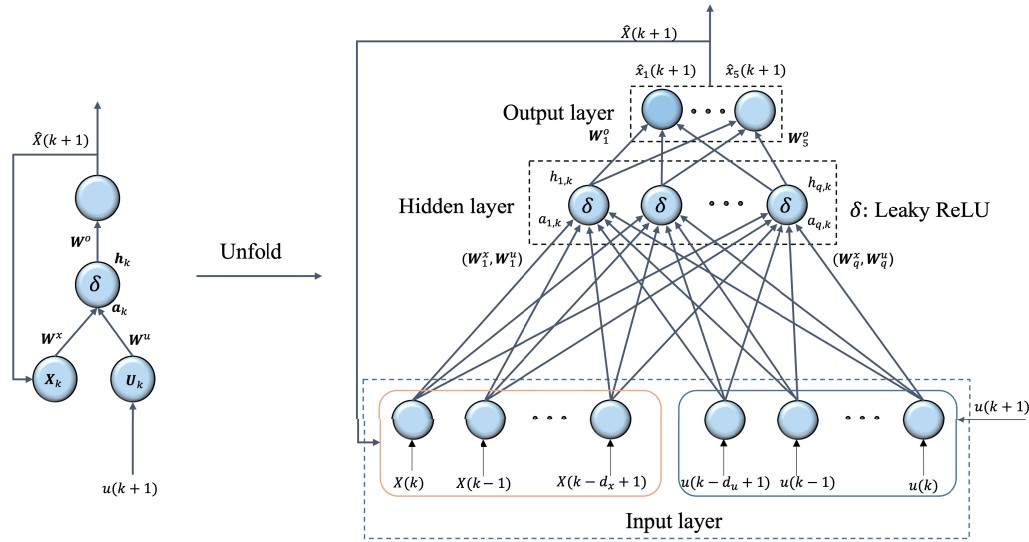

**Figure 2.** The ReLU-RNN architecture used to approximate system dynamic model.

From Figure 2, the input of the hidden neuron $a_{i,k}$ is computed by (13)

$$a_{i,k} = W_i^x X_k + W_i^u U_k + b_i,$$ (13)

where $W_i^x$ and $W_i^u$ are the weight vectors of the system state and control input for $i$th hidden neuron. $b_i$ signifies the bias of $i$th hidden neuron. The $i$th hidden neuron output $h_{i,k}$ is evaluated by (14)

$$h_{i,k} = \delta(a_{i,k}),$$ (14)

where $\delta(\cdot)$ represents the nonlinear activation function. We choose the leaky ReLU function as the nonlinear activation function, selected because it is useful for computing efficiently and preventing gradients from disappearing. The leaky ReLU function is detailed as (15)

$$\delta(\vartheta) = \begin{cases} \vartheta, & \text{if } \vartheta \geq 0, \\ 0.01\vartheta, & \text{if } \vartheta < 0, \end{cases}$$ (15)

where $\vartheta$ indicates the input variable of leaky ReLU function. Then, the predicted output is expressed as (16)

$$\hat{x}_j(k+1) = W_j^o h_k + b_j^o, \quad j = 1, 2, ..., 5,$$ (16)

where $W_j^o$ symbolizes the weight vector of $j$th output neuron concerning hidden layer neurons, and $b_j^o$ is the bias of $j$th output neuron. $h_k = [h_{1,k}, h_{2,k}, ..., h_{q,k}]$, where $q$ is the

number of hidden neurons. Combining (13)–(16), ReLU-RNN is capable of estimating system dynamic model by (17)

$$\hat{X}(k+1) = f_n(\boldsymbol{W}, \boldsymbol{b}, X_k, U_k), \tag{17}$$

where $\boldsymbol{W}$ and $\boldsymbol{b}$ represent the weights and bias of the NN, respectively.

Each batch of training data contains 1000 randomly selected state-input pairs throughout the training process. The state-input pairs are generated by the simulation of discretized system model (5). We choose the the mean squared error (MSE) as the loss function, which is denoted as (18)

$$\begin{aligned}
\ell(\hat{X}_k, X_k) &= \frac{1}{N} \sum_{j=1}^{N} ||\hat{X}(k+j) - X(k+j)||^2 \\
&= \frac{1}{N} \sum_{j=1}^{N} ||f_n(\boldsymbol{W}, \boldsymbol{b}, X_{k+j-1}, U_{k+j-1}) - X(k+j)||^2,
\end{aligned} \tag{18}$$

where $\hat{X}_k = \{\hat{X}(k+1), \hat{X}(k+2), ..., \hat{X}(k+N)\}$ symbolize the predicted values of system states. According to (18), the backpropagation method may be used to obtain the weight gradients $\frac{\ell(\hat{X}, X)}{\partial W}$ and bias gradients $\frac{\ell(\hat{X}, X)}{\partial b}$. Then, we adopt the Adam algorithm [59], a type of gradient descent method, to train the network. Using Intel(R) Core(TM) i7-8550U CPU, the learning rate is set to $1.0 \times 10^{-5}$, and the training process is completed after 50 min. The parameters of ReLU-RNN are set as follows. $d_x$ and $d_u$ are set to 5. The number of neurons in the hidden layer is 15, in the input layer is 30, and in the output layer is 5. The training findings are displayed in Section 4.

### 3.3. RNN and DEO Based NMPC Controller

In this subsection, we first introduce the DEO algorithm, which is based on the NMPC technique. Then, the RNN and DEO based NMPC controller is designed in detail.

The standard DEO is commonly indicated as DE/rand/1/bin [44]. A randomly selected population $\boldsymbol{P}$ consists of $N_P$ individuals corresponding to the prediction horizon of NMPC, each individual is an $N$-dimensional vector, which is represented by $U_i = [u_{i,1}, u_{i,2}, ..., u_{i,N}]$. The $U_i$ corresponds to the control input $U_{k+N}$ that will be optimized. The evolutionary generation time in DEO is expressed by $G = 0, 1, 2, ..., G_m$, where $G_m$ signifies the highest generation time. At $G$th generation, the $i$th individual of the $G$th generation population is designated as $U_i^G = [u_{i,1}^G, u_{i,2}^G, ..., u_{i,N}^G]$ with each element of $U_i^G$ constrained to $[u_L, u_U]$. $u_L$ and $u_U$ are the lower band and upper band of the control input, respectively. The population will vary with the evolution process, $\boldsymbol{P}^G$ stands for the $G$th generation population, and the initial population $\boldsymbol{P}^0$ is randomly generated with the boundary constraint $[u_L, u_U]$. The basic DEO algorithm operation procedure contains initialization, mutation, crossover, and selection, which are detailed as follows.

Initialization: To establish the initial point of the optimization search, the population needs to be initialized. Generally, one way to build an initial population is to randomly select from the values within a given boundary constraint. It is a common assumption that all populations with random initialization conform to a uniform probability distribution. Typically, each $j$th element of the $i$th individual in the $\boldsymbol{P}^0$ is initialized by (19)

$$u_{i,j}^0 = u_L + rand(0,1) \cdot (u_U - u_L), \quad (i = 1, 2, ....., N_P, j = 1, 2, ....., N), \tag{19}$$

where $rand(0,1)$ denotes a uniformly distributed random number in $[0,1]$.

Mutation: For each individual vector $U_i^G$, a mutant vector $V_i^G = [v_{i,1}^G, v_{i,2}^G, ..., v_{i,N}^G]$ at generation $G$ is generated by (20)

$$V_i^G = U_{r_1}^G + F \cdot (U_{r_2}^G - U_{r_3}^G), \quad r_1 \neq r_2 \neq r_3 \neq i, \quad (i = 1, 2, ....., N_P), \tag{20}$$

where $r_1, r_2, r_3 \in \{1, 2, 3, ..., N_P\}$ represent randomly chosen indices. $F \in [0, 2]$ is the zoom factor of the difference vector $(U_{r_2}^G - U_{r_3}^G)$. If the element $v_{i,j}^G$ of the mutant individual violates the feasible region boundary of the search space, a simple method to treatment this problem is to replace the element with a novel one formulated by Equation (19). Another method is boundary absorption, which is described as (21)

$$v_{i,j}^G = \begin{cases} u_L, & \text{if } v_{i,j}^G < u_L, \\ u_U, & \text{if } v_{i,j}^G > u_U, \end{cases} \quad (i = 1, 2, ....., N_P, j = 1, 2, ....., N), \tag{21}$$

The mainstream mutation strategies are described as follows (22)–(27)

(1)　DE/rand/1/bin

$$V_i^G = U_{r_1}^G + F \cdot (U_{r_1}^G - U_{r_2}^G), \quad r_1 \neq r_2 \neq r_3 \neq i, \tag{22}$$

(2)　DE/rand/2/bin

$$V_i^G = U_{r_1}^G + F \cdot (U_{r_2}^G - U_{r_3}^G) + F \cdot (U_{r_4}^G - U_{r_5}^G), \quad r_1 \neq r_2 \neq r_3 \neq r_4 \neq r_5 \neq i, \tag{23}$$

(3)　DE/best/1/bin

$$V_i^G = U_{best}^G + F \cdot (U_{r_1}^G - U_{r_2}^G), \quad r_1 \neq r_2 \neq i, \tag{24}$$

(4)　DE/best/2/bin

$$V_i^G = U_{best}^G + F \cdot (U_{r_1}^G - U_{r_2}^G) + F \cdot (U_{r_3}^G - U_{r_4}^G), \quad r_1 \neq r_2 \neq r_3 \neq r_4 \neq i, \tag{25}$$

(5)　DE/current-to-best/1/bin

$$V_i^G = U_i^G + F \cdot (U_{best}^G - U_{r_1}^G) + F \cdot (U_{r_2}^G - U_{r_3}^G), \quad r_1 \neq r_2 \neq r_3 \neq i, \tag{26}$$

(6)　DE/rand-to-best/1/bin

$$V_i^G = U_r^G + F \cdot (U_{best}^G - U_{r1}^G) + F \cdot (U_{r_2}^G - U_{r_3}^G), \quad r_1 \neq r_2 \neq r_3 \neq i, \tag{27}$$

where $r_1, r_2, r_3, r_4, r_5 \in \{1, 2, ..., N_P\}$ are randomly chosen individual indices. $U_{best}^G$ denotes the best fitness individual vector of $G$th generation.

Crossover: To maintain the diversity of the population, the crossover operation is introduced. Binomial crossover strategy is most frequently utilized, which is expressed as (28)

$$z_{i,j}^G = \begin{cases} v_{i,j}^G, & \text{if } rand(i) \leq C_r \quad or \quad j = randint(j), \\ u_{i,j}^G, & \text{otherwise}, \end{cases} \tag{28}$$

where $Z_i^G = [z_{i,1}, z_{i,2}, ..., z_{i,N}]$ stands for the trial vector. $C_r \in [0, 1]$ is the crossover rate, which determines how many elements are inherited from the mutant vector. $randint(j)$ is a randomly generated integer of $[1, N]$, which is used to make sure that at least one element of the trial vector is inherited from the mutant vector.

Selection: In the selection procedure, the fitness function of the DEO algorithm is designed according to the control objective. Since our goal is to execute position control of a single-link FJ robot, the cost function detailed in Equation (11) is designed as the fitness function (29) of the DEO algorithm.

$$f(U_i) = \alpha \sum_{j=0}^{N-1} \left[ \hat{x}_1(k+j+1) - x_1^{ref}(k+j+1) \right]^2 + \beta \sum_{j=0}^{N-1} (u_{i,j+1} - u_{i,j})^2. \tag{29}$$

The best individual in current population is selected by calculating the fitness function. The selection method is represented by (30)

$$U_i^{G+1} = \begin{cases} Z_i^G, & \text{if} \quad f(Z_i^G) < f(U_i^G), \\ U_i^G, & \text{otherwise,} \end{cases} \tag{30}$$

In this paper, we adopt DE/best/2/bin (25) as the mutation progress of the DEO algorithm. The adaptive mutation factor is applied to scale the difference vector, which is described as (31)

$$F = F_0 \times 2^\lambda, \quad \lambda = e^{1 - G_m/(G_m + 1 - G)}, \tag{31}$$

where $F_0$ signifies the initial mutation factor. The adaptive mutation factor is $2F_0$, which is a big value at the beginning of the evolution. Then, the diversity of individuals can be maintained, and it benefits for avoiding premature. In the later evolution period, the mutation rate is close to $F_0$, the better individual is retained, and the damage of the optimal solution is avoided. In this approach, the probability of searching for the global optimal solution is enhanced. The flow chart of the DEO algorithm is shown in Figure 3.

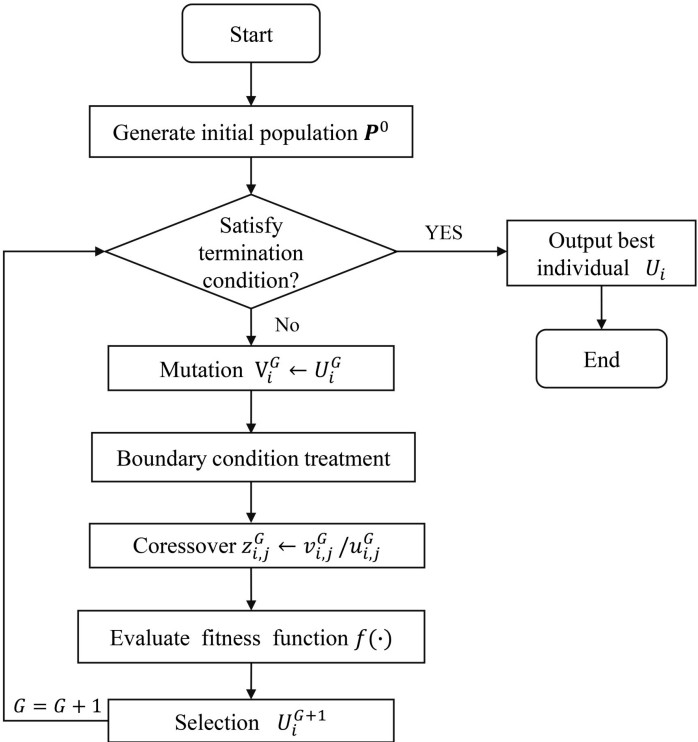

**Figure 3.** The flow chart of DEO algorithm.

The RNN and DEO based NMPC controller architecture is illustrated in Figure 4. First, we employ the ReLU-RNN described in Section 3.2 to approximate the system dynamic model. Then, the ReLU-RNN predictive model is employed for predicting system forward dynamics, which is capable of integrating into NMPC architecture for designing the NMPC controller. Finally, the DEO algorithm is utilized to optimize the control inputs, only the first term $u_0^\star$ is applied to the system, and the whole procedure runs as a closed-loop.

Based on the sampled system state information and the control inputs that will be optimized, the predicted position states $\hat{x}_1(k + j + 1)$ will be obtained via the ReLU-RNN predictive model (17). Then, the fitness function (29) is computed, and the system control inputs can be optimized via the DEO algorithm. The process of optimization via DEO algorithm is detailed in Algorithm 1.

---

**Algorithm 1** The optimization process of DEO.

---

**Input:**

    Individual dimension: $N$

    Maximum evolution generation: $G = 0, 1, 2, ..., G_m$

    Individual lower band: $u_L$

    Individual upper band: $u_U$

**Output:**

    The best $f(U_{best}^{G_m})$, and the best individual $U_{best}^{G_m}$

1: Initialize parameters: $C_r$, $F_0$, and $N_P$

2: Randomly initial population: $U^0 = [U_1^0, U_2^0, ..., U_{N_P}^0]$

3: **for** $G = 0$ to $G_m$ **do**

4:     Evaluate $f(U_i^G)$ and select the best individual $U_i^G$, $i = 1, 2, ..., N_P$

5:     Let $U_{best}^G \leftarrow U_i^G$

6:     Evaluate adaptive mutation factor: $F = F_0 \times 2^\lambda$, $\lambda = e^{1 - G_m/(G_m + 1 - G)}$

7:     **for** $i = 1$ to $N_p$ **do**

8:         Randomly generate: $r_1, r_2, r_3, r_4$ and $r_1 \neq r_2 \neq r_3 \neq r_4 \neq i$

9:         $V_i^G = U_{best}^G + F \cdot (U_{r_1}^G - U_{r_2}^G) + F \cdot (U_{r_3}^G - U_{r_4}^G)$

10:        Randomly generate $randint(j)$, $randint(j) \in \{1, 2, ..., N\}$

11:        **for** $j = 1$ to $N$ **do**

12:            **if** $rand(0, 1) < CR$ or $j = randint(j)$ **then**

13:              $z_{i,j}^G \leftarrow v_{i,j}^G$

14:            **else**

15:              $z_{i,j}^G \leftarrow u_{i,j}^G$

16:            **end if**

17:            **if** $z_{i,j}^G \leq u_L$ **then**

18:              $z_{i,j}^G \leftarrow u_L$

19:            **end if**

20:            **if** $z_{i,j}^G \geq u_U$ **then**

21:              $z_{i,j}^G \leftarrow u_U$

22:            **end if**

23:        **end for**

24:        **if** $f(Z_i^G) \leq f(U_i^G)$ **then**

25:            $U_i^G \leftarrow Z_i^G$

26:        **end if**

27:     **end for**

28: **end for**

29: **return** The best $f(U_{best}^{G_m})$, and the best individual $U_{best}^{G_m}$.

---

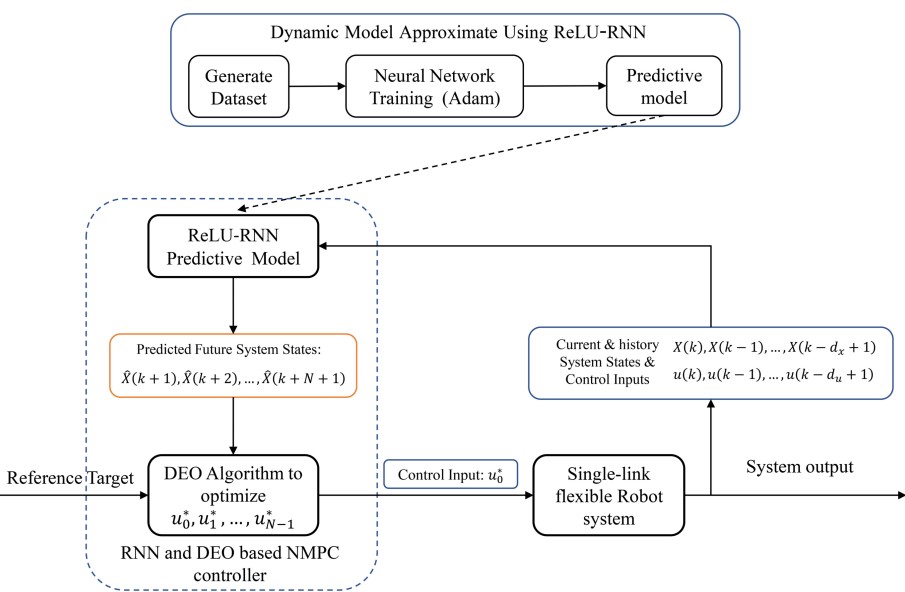

**Figure 4.** The architecture of RNN and DEO based NMPC controller.

A corresponding summary of the RNN and DEO based NMPC scheme can be presented as follows:

Step1. Obtaining the current system states and the saved history system states information along with system control inputs from the single-link flexible joint robot system.

Step2. Based on the system state information, using the ReLU-RNN predictve model to predict future position states with $N$ time steps.

Step3. According to the predicted system state information and the designed cost function, using the DEO (Algorithm 1) to solve the NMPC controller.

Step4. Applying the first term ($u_0^\star$) of the optimized control inputs to the system until the next time step.

Step5. Time step proceeds forward one step ($k = k + 1$). Then, it updates the saved history system state information, and returns to Step 1.

### 3.4. Control Stability Analysis

The NMPC is obtained for the plant (9) by minimizing the cost function (11) satisfy the terminal constraint (12). It is clearly that $J(\boldsymbol{X}(k), U(k)) \geq 0$ and $J(\boldsymbol{X}(k), U(k)) = 0$ only if $U(k) = 0$, and $J(\boldsymbol{X}(k), U(k))$ is decrescent. We assume that $\boldsymbol{X}(k) = 0$ and $U(k) = 0$ is an equilibrium condition for the plant: $0 = f(0,0)$. The MPC control law is $U^\star(k) = [u_0^\star, u_1^\star, ..., u_{N-1}^\star]$. Thus, the equilibrium point $\boldsymbol{X}(k) = 0$ and $U(k) = 0$ is stable, providing that the optimization problem is feasible and is solved at each time step [23,60,61].

We define $J^\star(\boldsymbol{X}(k), U^\star(k))$ as the optimal value of $J(\boldsymbol{X}(k), U^\star(k))$ which corresponds to the optimal control input $U^\star(k)$. It is clearly that $J^\star(\boldsymbol{X}(k+1), U^\star(k+1)) \geq 0$, and $J^\star(\boldsymbol{X}(k), U^\star(k)) = 0$ only if $U^\star(k) = 0$. We will show that $J(\boldsymbol{X}(k+1), U^\star(k+1)) \leq J^\star(\boldsymbol{X}(k), U^\star(k))$, and hence that $J^\star(\boldsymbol{X}(k), U^\star(k))$ is a Lyapunov function for the closed-loop system.

As usual in stability proofs, we will assume that the ReLU-RNN predicitve model is perfect, so that the predicted and real state trajectories coincide: $X(k+j) = \hat{X}(k+j)$ if $u(k+j) = u^\star(k+i)$.

Let define

$$J(\boldsymbol{X}(k), U(k)) = \min_U \sum_{j=0}^{N-1} G(\boldsymbol{X}(k), U(k)) \tag{32}$$

where

$$G(\boldsymbol{X}(k), U(k)) = \alpha \left[ x_1(k+j+1) - x_1^{ref}(k+j+1) \right]^2 + \beta [u(k+j+1) - u(k+j)]^2 \quad (33)$$

With this assumption we have

$$
\begin{aligned}
J^\star(\boldsymbol{X}(k+1), U(k+1)) &= \min_U \sum_{j=1}^N G(\boldsymbol{X}(k+j+1), U(k+j)) \\
&= \min_U \sum_{j=1}^N G(\boldsymbol{X}(k+j), U(k+j-1)) - G(\boldsymbol{X}(k+1), U(k)) + G(\boldsymbol{X}(k+N+1), U(k+N)) \\
&\le -G(\boldsymbol{X}(k+1), U(k)) + J^\star(\boldsymbol{X}(k), U(k)) + \min_U \{ G(\boldsymbol{X}(k+N+1), U(k+N)) \}.
\end{aligned}
\quad (34)
$$

We have assumed that the terminal constraint is satisfied, the optimization problem was assumed to be feasible, so we can make $U(k+N) = 0$ and stay at $\boldsymbol{X}(k) = 0$, which gives

$$\min_U \{ G(\boldsymbol{X}(k+N+1), U(k+N)) \} = 0. \quad (35)$$

Since $G(\boldsymbol{X}(k), U(k)) \ge 0$, we can conclude that $J^\star(\boldsymbol{X}(k+1), U(k+1)) \le J^\star(\boldsymbol{X}(k), U(k))$. Thus, $J^\star(\boldsymbol{X}(k), U(k))$ is a Lyapunov function, and we conclude by Lyapunov's theorem that the equilibrium $\boldsymbol{X}(k) = 0$, $U(k) = 0$ is stable.

## 4. Numerical Simulations

In this section, MATLAB 2019b is employed to create numerical simulations to demonstrate the effectiveness and performance of our suggested method. To verify the superiority of this method, the conventional PD and DDP MPC methods were considered comparatives. DDP MPC is an NMPC technique that uses DDP to solve the MPC controller. To achieve a fair comparison, the predictive model and cost function used in DDP MPC were the same as the RNN and DEO based NMPC method, and the parameters of the controllers were carefully adjusted.

The model approximated by ReLU-RNN, which has been described in Section 3.1, was used for designing the NMPC controller. The discretized system dynamic model (5) was simulated in MATLAB, which is regarded as the real system platform. Table 1 lists the parameters of the simulated model.

**Table 1.** The simulation model parameters of single-link FJ robot.

| Parameters | Values | Parameters | Values |
|---|---|---|---|
| $J_1$ | 0.8 kg· m$^2$ | $R$ | 5.3 $\Omega$ |
| $J_2$ | 0.1 kg· m$^2$ | $K_{f_1}$ | 2.0 |
| $N$ | 200 | $K_{f_2}$ | 2.0 |
| $K$ | 70 Nm/rad | $m$ | 0.3 kg |
| $L$ | $1.4 \times 10^{-5}$ H | $l$ | 0.5 m |
| $K_\tau$ | $9.3 \times 10^{-3}$ Nm/A | $g$ | 9.8 m/s$^2$ |
| $K_e$ | 0.1 V/rad/s | - | - |

The MSE (18) was employed to evaluate the performance of the approximated model. Table 2 shows the model approximation results. The findings revealed that the prediction precision of the learned model was relatively accurate.

**Table 2.** The MSE of the ReLU-RNN predictive model.

| States | $x_1$ (rad) | $x_2$ (rad/s) | $x_3$ (rad) | $x_4$ (rad/s) | $x_5$ (A) |
|---|---|---|---|---|---|
| MSE | $3.14 \times 10^{-7}$ | $4.76 \times 10^{-7}$ | $2.60 \times 10^{-7}$ | $1.44 \times 10^{-7}$ | $6.25 \times 10^{-8}$ |

Figure 5 displays the progress of the multi-step prediction of the ReLU-RNN predictive model. Correspondingly, Figure 6 shows the absolute errors. The figures show that, even if a forward prediction was 20 time steps, the performance of the ReLU-RNN predictive model was also satisfied, and it could be used to establish an NMPC controller.

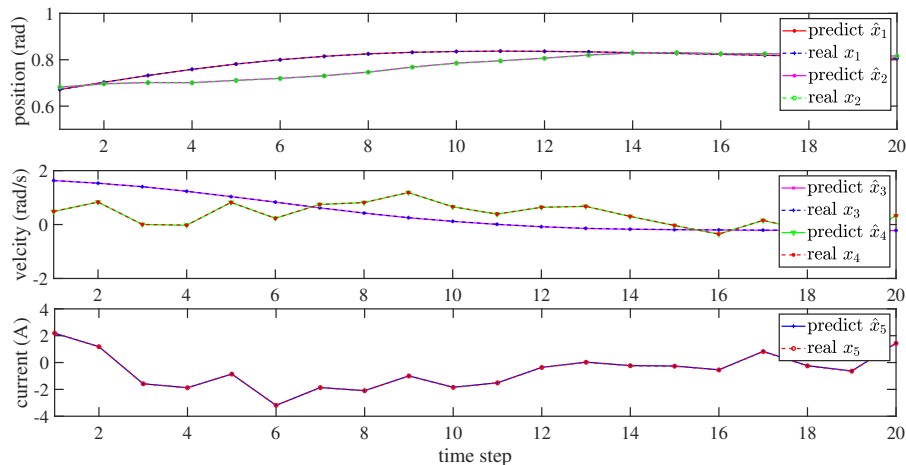

**Figure 5.** The progress of multi-step prediction.

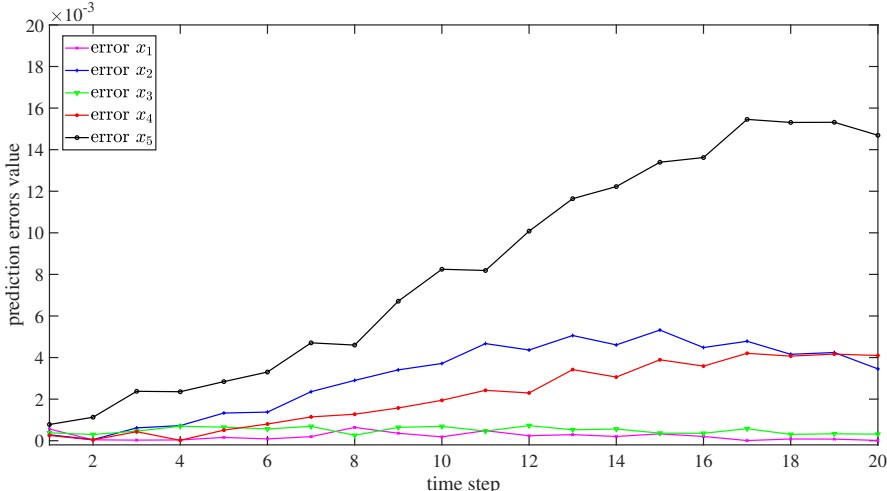

**Figure 6.** The absolute errors of multi-step prediction.

In the simulation procedure, the time step for the suggested method and DDP MPC was 20 ms, and the prediction horizon was five time steps. The parameters of the DEO algorithm are displayed as follows, $F_0 = 0.5$, $CR = 0.5$, $NP = 30$, $G_m = 200$. The control inputs were constrained in $[-24 \text{ V}, 24 \text{ V}]$.

The experimental results proposed by [44,62] have shown that the DEO has good convergence properties. To demonstrate the convergence of DEO, the cost values of the optimization process are plotted in Figures 7 and 8. Figure 7 displays the cost values in evolutionary iteration at each time step. It can be seen that the cost value converged to a fixed value after 80 iterations. Figure 8 shows the optimized cost values at the target tracking process. We can see that the cost values converged to a small value with the increase of time step, which indicates that the DEO could solve the proposed controller effectively.

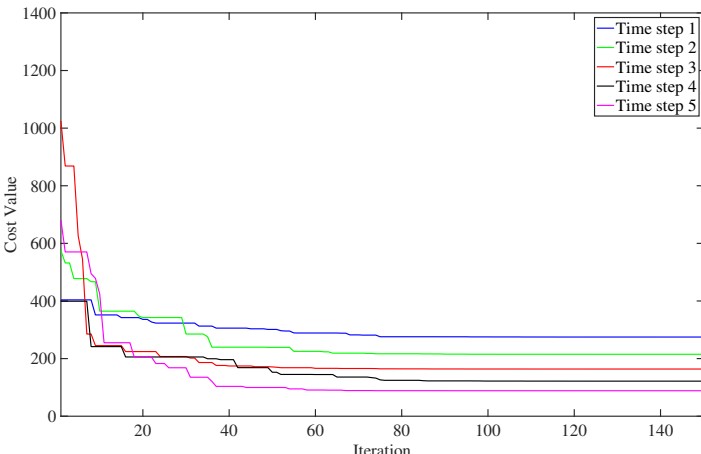

**Figure 7.** The cost values of the optimization process at five adjacent time steps.

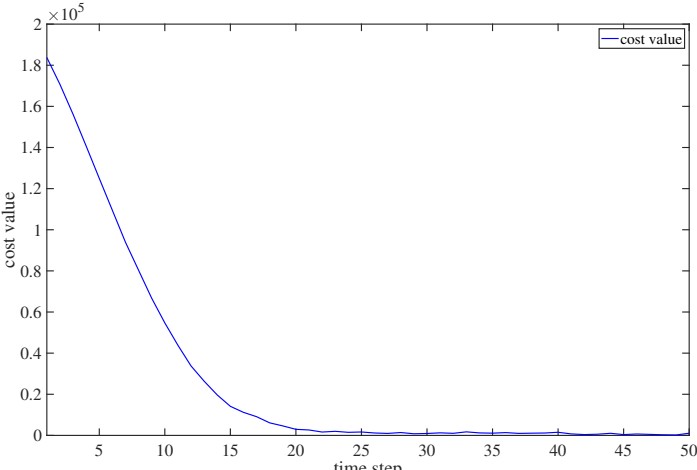

**Figure 8.** The cost values of the optimization process at each time step.

Figures 9 and 10 depicts the tracking performance of different controllers, while Table 3 indicates the state error of different controllers. The findings illustrate that both controllers could efficiently control the system.

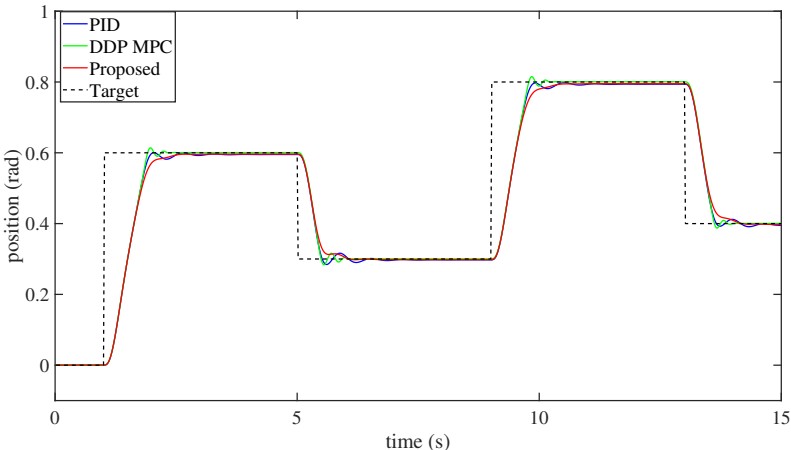

**Figure 9.** The target tracking process of different controllers.

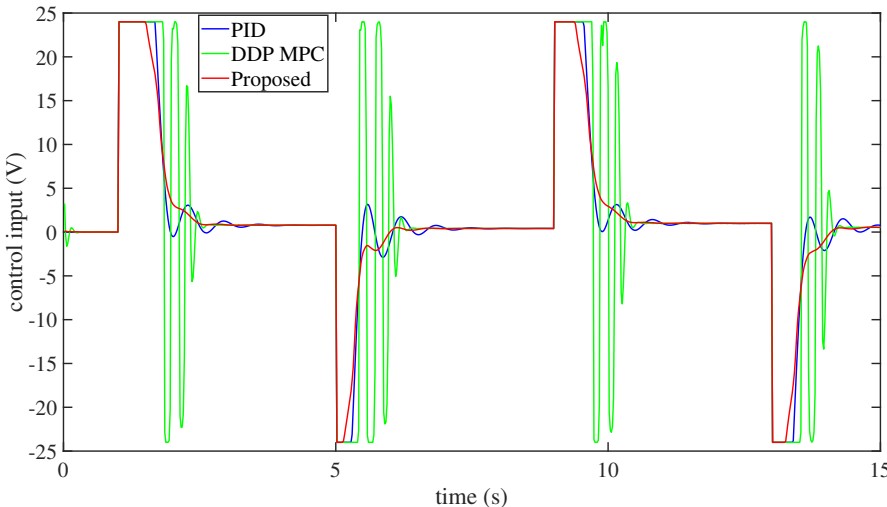

**Figure 10.** The control actions of different controllers.

**Table 3.** Comparison of the state error of different controllers.

| Controller | Proposed | PID | DDP MPC |
|---|---|---|---|
| Proposed | 0.0037 | 0.0049 | 0.0016 |

Figure 9 indicates that there were some overshoots and residual vibration in the system response when controlled by the PD and DDP MPC methods. This is due to the existence of an elastic element in the FJ robot, which led to the overshoots and residual vibration being easily inspired. Nevertheless, from Figure 9, we can see that the proposed controller was able to reduce the overshoots and suppress the residual vibration.

Table 3 demonstrates that our controller had a certain degree of precision control, and the precision was better than the PD controller. The DDP MPC controller achieved higher precision than our controller, but a closer look at the tracking progress in Figure 9, shows that the tracking process of our controller was smooth, with few overshoots and the vibration was well suppressed. Figure 10 depicts the controller actions. The control signal of the DDP MPC controller fluctuated greatly, the PD controller presented smaller fluctuations, and the proposed controller had the smallest fluctuations. The fluctuations in the controller signal had a great influence on the system, potentially reducing the service life of the robot and even leading to mechanical damage. The influence of controller signal fluctuations was, to some extent, more essential than control precision. It indicates that our strategy was more suitable for FJ robot control.

The need for a closed-loop system is important in the presence of external disturbances. To verify that the proposed controller is robust to external disturbances, we added external disturbances to the system. Figures 11 and 12 show the system responses with external disturbances. As can be seen from Figure 11, the system responded quickly and remained stable. The control performance was also fairly satisfactory. Figure 12 depicts the control actions, which demonstrates that the proposed controller could be solved by the DEO efficiently and it could achieve a good robustness against external disturbances.

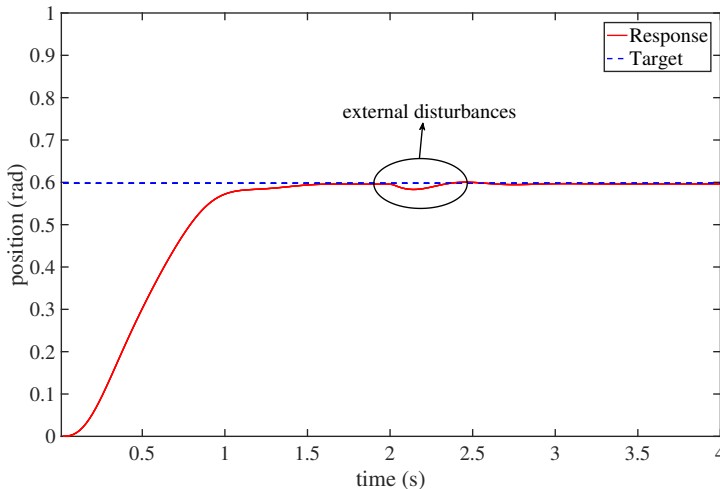

**Figure 11.** The target tracking process with external disturbances.

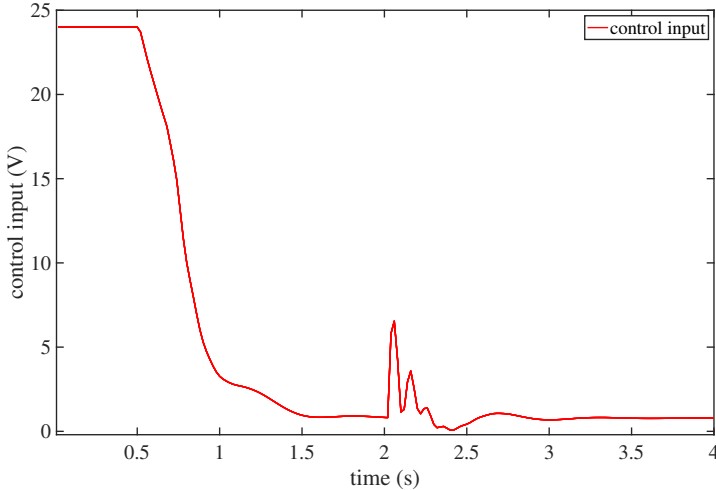

**Figure 12.** The control actions with external disturbances.

Based on the above investigation, we conclude that the performance of the RNN and DEO based NMPC method was better than that of the PD and DDP MPC methods. In addition, this method achieved a good robustness against external disturbances. The merit of the proposed method was that not only was the control precision satisfied, but also the overshoots and residual variation were suppressed well.

## 5. Conclusions

This work presents an RNN and DEO based NMPC approach for position control of a single-link FJ robot. First, the system dynamic model has been approximated using a simple three-layer ReLU-RNN. Then, according to the RNN predictive model and MPC method, the RNN and DEO based NMPC controller was designed, in which the DEO algorithm was utilized to optimize the control inputs. Finally, through comparative numerical simulations, the effectiveness and performance of the proposed technique have been verified. The simulation findings have shown that the suggested method is superior to that of the PD and DDP MPC methods, which is capable of minimizing overshoots and suppressing residual variation with the control precision satisfied.

The parallel DEO can speed up the optimization process because DEO is a stochastic optimization algorithm that is inherently parallel. In the future, considering the optimization solution time, we will evaluate the RNN and parallel DEO based NMPC approach that can be utilized for implementing real-time NMPC, and it will be further verified

by experiments. In addition, we intend to apply it to multi-degree-of-freedom FJ robot applications.

**Author Contributions:** conceptualization, A.Z., Z.L., B.W. and Z.H.; methodology, A.Z.; software, A.Z.; validation, A.Z., Z.L., B.W. and Z.H.; formal analysis, A.Z.; investigation, A.Z.; resources, A.Z., Z.L., B.W. and Z.H.; writing—original draft preparation, A.Z.; writing—review and editing, A.Z.; visualization, A.Z.; supervision, Z.L., B.W. and Z.H.; project administration, A.Z. and Z.L.; funding acquisition, Z.L. All authors have read and agreed to the published version of the manuscript.

**Funding:** This research received no external funding.

**Conflicts of Interest:** The authors declare no conflict of interest.

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
