# Peer review of "Nonlinear Model Predictive Control of Single-Link Flexible-Joint Robot Using Recurrent Neural Network and Differential Evolution Optimization"

_electronics, doi:10.3390/electronics10192426_

Round 1
Reviewer 1 Report
ID: electronics-1379400-peer-review-v1
Title: Nonlinear Model Predictive Control of Single-link Flexible-Joint Robot Using Recurrent Neural Network and Differential Evolution Optimization
The focus of the paper is on the use of a recurrent neural network (RNN) and differential evolution optimization (DEO) based nonlinear model predictive control (MPC) technique to control the position of a single link flexible-joint (FJ) robot. Authors compared numerical simulation findings to demonstrate the efficiency of the proposed approach.
In general, the paper has a good quality. The topic can be interest of robotic researchers. However, I have some comments about organization and content of the paper. Please see the following comments:
First, I do not know if the journal accepts the graphical abstract or not. However, I think it is a good idea that authors develop a graphical abstract based on Figure 4, which shows the architecture of RNN and DEO based nonlinear MPC controller as a high-level presentation of approaches. It would be nice if it has something about FJ robots even if it be brief.
My main concern is about the methodologies of the paper. The MPC is one of them, but I think there is not enough discussion about the challenges of its implementation in this paper. Part of the challenge in implementing MPC is that the regulatory control layer is not often a given (or should not be taken as a given). The design problem is really one of deciding on the best overall structure for the regulatory level and MPC, given the control objectives, expected constraints, at least qualitative knowledge of the expected disturbances, and robustness considerations. Similarly, the selection of the controlled variables for MPC is not one of simply deciding which subset of available measurements should be selected.
An additional challenge is that it is common for an MPC controller to contain logic to switch off if an integrating variable cannot be balanced (zero difference) at steady state, thus making integrating variables more sensitive.
As mentioned at Pages 5 and 10, the optimization procedure is repeated at each time step, which runs as a closed-loop. A challenge with closed-loop identification is the importance of obtaining an accurate noise model, which is problematic in practice, since typical process disturbances cannot be captured by white noise, passed thru a linear filter. So, I believe that authors should clarify it further. In practice, one can attempt to minimize the bias by “overwhelming” noise feedback in the frequency range of interest.
The challenges of the optimization of robotic systems should also be discussed. The collision/deadlock avoidance is studied in robots. For the sake of generality, my suggestion is that authors refer to three important studies in this area. [a] deadlock-free scheduling of manufacturing systems using petri nets and dynamic programming, IFAC Proceedings Volumes, vol.32, pp. 4870-4875 [b] deadlock prevention and performance oriented supervision in flexible manufacturing cells: a hierarchical approach, Robotics and Computer-Integrated Manufacturing, vol.27, pp. 591-603 [c] resolution of deadlocks in a robotic cell scheduling problem with post-process inspection system: avoidance and recovery scenarios, 2015 IEEE International Conference on Industrial Engineering and Engineering Management (IEEM), 2015, pp. 1107-1111
Other errors:
Page 6: We choose the the mean --> We choose the mean
Page 10: Table 1: The simulation --> Table 1. The simulation
…
Reviewer 2 Report
The article employ modern approaches from control theory to a mechanical drivetrain with an elastic link. To call this "a robot" you need to specify what kind of work that mechanical system can perform in practice.
The article can be published in this form because it is well structured and scientifically consistent. However we should be clear that this is not a high quality research due to following points:
1. System is investigated only in simulation. Experimental evaluation of such dynamic optimization system is obligatory due to possible convergence problems in real-time and poor robustness. Hence, the system should prove that overperform PID and classical MPC in experiments.
2. Proof of convergence of optimization is missing or strategy what the system would do if optimization converge to infeasible solution.
3. What is the benefit of neural network model over nonlinear model based on analytical mechanics. Such advantages should be proved with specific tests from system identification theory.
4. Performance difference between proposed controller and other classical approaches like PID is too small (that is because the mechanical system chosen is too simplistic). So for this particular system it is clear for a control system engineer that PID can be tuned to perform no worse than proposed nonlinear controller.
5. Proof of stability of the closed loop system with the proposed nonlinear controller is missing. As well as a proof of robustness.
Round 2
Reviewer 1 Report
The paper ID:electronics-1379400-peer-review-v2
In this revision, RNN and MPC methods and FJ robots are better explained. The graphical abstract is also helpful, and Figure 2 has a better presentation. Finally, the referencing format has consistency in the current format.
I have looked at highlighted paragraphs and I can see that authors significantly improved the quality of paper. So, it can be accepted as it is.